# The Role of Anti-SSB/La Antibodies as Predictors of Decreased Diffusing Capacity of the Lungs for Carbon Monoxide (DLCO) in Primary Sjögren Disease

**DOI:** 10.3390/ijms26125867

**Published:** 2025-06-19

**Authors:** Simona Caraiola, Laura Voicu, Daniela Opriș-Belinski, Claudia Oana Cobilinschi, Magda Ileana Pârvu, Ion Andrei Ion, Daniela Ștefana Gologanu, Răzvan Adrian Ionescu

**Affiliations:** 1Fifth Department-Internal Medicine (Cardiology, Gastroenterology, Hepatology, Rheumatology, Geriatrics), Family Medicine, Occupational Medicine, Faculty of Medicine, “Carol Davila” University of Medicine and Pharmacy, 050474 Bucharest, Romania; 2Internal Medicine Department, Colentina Clinical Hospital, 020125 Bucharest, Romania; 3“Prof. Dr. C.C. Iliescu” Emergency Institute for Cardiovascular Diseases, 022328 Bucharest, Romania; 4Department of Rheumatology and Internal Medicine, Sfânta Maria Clinical Hospital, 011172 Bucharest, Romania; 5Rheumatology Department, Colentina Clinical Hospital, 020125 Bucharest, Romania; 6Pneumology Department, Colentina Clinical Hospital, 020125 Bucharest, Romania

**Keywords:** primary Sjögren’s Disease, interstitial lung disease, diffusing capacity of the lungs for carbon monoxide, anti-SSA/Ro antibodies, anti-SSB/Ro antibodies

## Abstract

Lung involvement is the most common extraglandular manifestation of primary Sjögren’s Disease (pSjD). There is an increasing interest in finding the clinical/serological risk predictors of this feature. A cross-sectional study evaluating anti-SSA/Ro antibodies, anti-SSB/La antibodies, rheumatoid factor, antinuclear antibodies, and the diffusing capacity of the lungs for carbon monoxide (DLCO) in 26 pSjD patients who presented interstitial changes on the chest CT scan was performed. The titres and positivity rates for anti-SSA/Ro (*p* = 0.02, *p* = 0.02) and anti-SSB/La antibodies (*p* = 0.01, *p* = 0.001) proved to be significantly increased in patients with abnormal DLCO. Anti-SSB/La antibodies’ titres seemed to be the best predictor for decreased DLCO–AUC 0.791 (0.587–0.994), *p* = 0.016. A close-to-significance decrease was found in the titres (*p* = 0.07) and positivity rates—*p* = 0.09 and OR of 0.15 (0.01–1.63)—of anti-SSB/La antibodies in patients with usual interstitial pneumonia (UIP), indicating their possible protective role against UIP. The lymphocytic interstitial pneumonitis (LIP) pattern on lung CT scan was significantly associated with the simultaneous positivity of the four examined serological markers (*p* = 0.03). The increase in anti-SSB/La antibody positivity rate in patients with LIP patterns was situated close to the significance level (*p* = 0.09). Quadruple positivity, as well as isolated anti-SSB/La positivity, could be risk factors for developing LIP in pSjD patients. Thus, anti-SSB/La antibodies might represent a marker of lung involvement in pSjD patients.

## 1. Introduction

Sjögren’s Disease (SjD) is an autoimmune disorder with chronic evolution that affects the exocrine glands, additionally generating systemic manifestations [1,2,3]. In primary SjD (pSjD), the defining features of the disease appear isolated, while in the case of secondary SjD, they are associated with the presence of a distinct connective tissue disease [4].

The classification criteria for pSjD were revised in 2016 by the American College of Rheumatology and the European League Against Rheumatism [5]. The histopathologic examination of a labial salivary gland, the evaluation of the Ocular Staining Score, the measurement of unstimulated whole saliva flow, the detection of anti-SSA/Ro antibody positivity, and the quantification of tear secretion by Schirmer’s test are the recommended instruments to be used in the classification process [5]. Anti-SSB/La antibodies, rheumatoid factors (RFs), and antinuclear antibodies (ANAs) that were found among the old classification criteria [6] are not present in the revised variant [5].

SjD can generate heterogeneous extraglandular impairment, from musculoskeletal and cutaneous manifestations to cardiovascular, hematologic, pulmonary, neurological, renal, or gastrointestinal disorders [7].

Due to different methods of defining pulmonary involvement (clinical symptoms associated with abnormal pulmonary function tests and/or abnormal findings upon high-resolution computed tomography—HRCT), the prevalence of pulmonary involvement in SjD has large variabilities between 9 and 22% [8] and a systematic review described 17% prevalence of interstitial lung disease (ILD) in pSjD [9].

Pulmonary manifestations are varied, including most often ILD and, in a minority of cases, obstructive airway disease [8,10].

Clinical symptoms are generally nonspecific, such as dry cough and exertional dyspnea [11]. However, the disease can frequently remain in a subclinical state [12]. It may also be difficult to detect pulmonary involvement by symptoms, as dyspnea may be due to fatigue, anemia, and joint and muscular damage [8], and coughing may be due to the dryness of mucosae (xerotrachea).

ILD diagnosis implies clinical and imagistic evaluation, pulmonary function tests, and histopathologic examinations [13,14].

The main radiological ILD patterns identified by HRCT of the chest in pSjD patients are non-specific interstitial pneumonitis (NSIP), usual interstitial pneumonia (UIP), and lymphocytic interstitial pneumonitis (LIP) [9,11]. In the case of ILD, pulmonary function tests reveal restrictive ventilatory dysfunction (low forced vital capacity—FVC; low total lung capacity—TLC; and normal forced expiratory volume in one second—FEV1/FVC ratio) and/or the low diffusing capacity of the lung for carbon monoxide (DLCO). The latter may be an early sign of interstitial disease [15].

As ILD represents an important mortality cause in these patients [16], recent ACR/CHEST consensus guidelines indicate recommendations for the screening and monitoring of ILD in patients with systemic autoimmune rheumatic disease, including pSjD [17].

In recent years, significant research efforts have been made in order to determine the possible predictors of lung impairment in SS. The results have brought forward various clinical and para-clinical factors, such as older age at diagnosis, male gender, smoking history, respiratory symptoms (dry cough and dyspnea), Raynaud’s phenomenon, high levels of C-reactive protein, elevated erythrocyte sedimentation rates, elevated lactate dehydrogenase, low levels of albumin, serum levels of IgG and IgM, and other serologic markers that seem to be connected to lung involvement in SjD [9,18,19,20,21,22,23]. Moreover, recent findings suggest that the presence of anti-Ro-52 antibodies is associated with an increase in the risk of ILD in pSjD patients [24]. Data regarding the association of other serological markers, such as anti-SSA/Ro antibodies or anti-SSB/La antibodies, with ILD development are scarce and require further research.

The present study aimed to investigate the predictive role of different clinical and serological factors, including anti-SSA/Ro and anti-SSB/La antibodies, in the decrease in the DLCO of patients with pSjD-ILD and also to describe patient characteristics in different radiological (HRCT) patterns.

## 2. Results

The study group comprised 26 patients in whom the lung CT scan result indicated interstitial changes. More than 80% (22) of them were females. A number of 17 (65%) patients registered abnormal (low) DLCO values. The mean age at inclusion in the study population was 57 years, while the mean age at the onset of the disease was 51 years. The mean duration of the rheumatologic disease at the time of the evaluation was seven years (Table 1). Only 3 of the 26 (11%) examined patients were smokers.

More than 90% (24) of the patients had positive ANA. Over 80% (22) of them had positive anti-SSA/Ro antibodies, while more than 50% (14) had positive anti-SSB/La antibodies. RF was found to be positive in over half (14, 53%) of the enrolled patients as well. When examining the titres, high values were noted for anti-SSA/Ro antibodies, anti-SSB/La antibodies, and ANA in the study population. Eight of the patients were positive for all four examined serological markers (ANA, anti-SSA/Ro antibodies, anti-SSB/La antibodies, and RF).

In about 60% (15) of the cases, high-resolution CT scan images revealed an NSIP aspect. Six patients (23%) had an LIP radiological pattern, and the remaining five (19%) were classified as having UIP characteristics.

Cryoglobulinemia was found in three (11%) of the subjects. Nine patients (34%) presented hypergammaglobulinemia, whilst hypocomplementemia was detected in ten (38%) of them. Extraglandular involvement (excepting ILD) was detected in a large majority of the study group (24 patients, 92%). About 70% (19) of the patients had hematologic involvement. Neurologic manifestations of the disease were recognized in nine cases (34%). Renal involvement was found in seven (26%) of the enrolled patients, whereas only three (11%) patients had adenopathies.

When comparing patients with abnormal DLCO to those who registered normal DLCO values (Table 2), the former proved to have a significantly higher age, both at enrolment (*p* = 0.02) and at the onset of the disease (*p* = 0.03). Moreover, the titres and rate of positivity for anti-SSA/Ro (*p* = 0.02 and *p* = 0.02, respectively) and anti-SSB/La (*p* = 0.01 and *p* = 0.001, respectively) antibodies in these patients were significantly superior to those of the subjects who had no changes in the DLCO measurement. A significant reduction in the values of TLC was noted in the abnormal DLCO subgroup (*p* = 0.01). When comparing normal and abnormal TLC between patients with decreased DLCO or DLCO within the normal range, no significant differences were registered, since 23 out of the 26 evaluated patients had normal TLC. In addition, the presence of hypocomplementemia was found to be close to the significance level (*p* = 0.05).

However, when performing multivariate analysis, statistical significance was maintained only for age at enrolment (*p* = 0.000), age at onset of the rheumatologic disease (*p* = 0.017), anti-SSA/Ro positivity (*p* = 0.000), and anti-SSB/La positivity (*p* = 0.000) (Table 3).

When performing ROC analysis for evaluating the utility of anti-SSA/Ro antibodies, anti-SSB/La antibodies, and ANA titres in predicting a low DLCO in pSjD patients in whom CT scan images revealed lung involvement, the most important AUC was registered for anti-SSB/La antibodies—AUC 0.791 (0.587–0.994); *p* = 0.016 (Table 4).

Patients who presented NSIP radiological patterns were compared with the rest of the enrolled subjects. Renal involvement proved to be significantly associated with the NSIP aspect (*p* = 0.03) (Table 5).

In patients classified as UIP according to their lung CT scan aspect, male gender, titres, and the rate of positivity of anti-SSB/La antibodies (lower titres and lower rate of positivity in patients having UIP) were found to be close to the significance level (Table 5).

In the subgroup of LIP patients, a significant association was found between this radiological pattern and the presence of quadruple positivity (*p* = 0.03), while the *p*-value for anti-SSB/La antibody positivity approached the level of significance (Table 5).

Extraglandular involvement (excepting ILD) was found in all patients who registered abnormal DLCO values. Furthermore, all of them had a positive ANA and positive anti-SSA/Ro antibodies.

Quadruple positivity proved to be significantly more frequent among patients with abnormal DLCO and LIP radiological patterns (*p* = 0.03) (Table 6).

A significantly higher age at enrolment (*p* = 0.01), higher TLC (*p* = 0.02), and lower anti-SSA/Ro antibody titres (*p* = 0.04) were detected in patients with abnormal DLCO and UIP findings upon the chest CT scan result. Concurrently, *p*-values corresponding to anti-SSB/La antibody positivity registered an extremely close-to-significance value (Table 6).

Near-significance *p*-values were found for renal impairment and hypergammaglobulinemia in patients with abnormal DLCO and NSIP findings in their high-resolution CT images (Table 6).

## 3. Discussion

Being the most common extraglandular manifestation of SjD [23], the lung involvement registered in these patients has been extensively studied in the last decade. NSIP seems to be the most prevalent radiological pattern, being present in about half of the ILD cases in pSjD patients. UIP and LIP have registered similar percentages for around 15% of the cases [16]. The distribution of ILD subtypes identified in our study group was comparable to the values found in the literature—57% NSIP, 23% LIP, and 19% UIP.

Patients with pSjD-ILD report impaired health related to quality of life and a higher risk of death, suggesting the importance of the early diagnosis and treatment of this type of pulmonary involvement [8].

A clear emphasis has been placed on the detection of clinical or serological factors lately, which could be used as risk predictors of lung impairment in SjD. Several studies concur that older SjD patients, as well as patients diagnosed with SjD at a more advanced age, present a higher risk of developing ILD [20,25,26,27,28]. In addition, recent results indicate a significant reduction in DLCO in pSjD patients with ILD [29]. Although all patients included in the present research had interstitial changes on the lung CT scan images, our results complement previous studies, since both the age at enrolment (*p* = 0.02) and the age at onset of disease (*p* = 0.03) were significantly higher in patients with low DLCO values. Male gender has also been mentioned as an ILD risk factor in the literature [25,26,27]. Nevertheless, no significant difference in the DLCO values could be found between males and females in the present study, most probably due to the reduced number of male patients that were included (only 15% of the cases).

Prior research identified a negative correlation between the TLC and the severity of CT scan changes in pSjD patients with ILD [23]. It is important to mention that most of the patients with decreased DLCO had normal TLC. Thus, DLCO might be more sensitive than TLC in detecting ILD among SjD patients; therefore, DLCO could be an early sign of and a screening tool for ILD.

It is a known fact that the presence of anti-SSA/Ro or anti-Ro52 antibodies is associated with ILD in pSjD patients [24,30]. Anti-SSA/Ro positivity has been linked to a significant decrease in DLCO in pSjD patients when compared to anti-SSA/Ro-negative patients [31]. Additionally, anti-SSA/Ro, anti-SSB/La, or anti-Ro52 positivity was also linked to higher ILD prevalence in other disorders, such as inflammatory myopathies [32]. Moreover, Yazisiz et al. have identified anti-SSA/Ro antibodies, anti-SSB/La antibodies, and RF as possible predictors for lung impairment in pSjD [33]. Our results are consistent with these findings. Both the titres and positivity rates for anti-SSA/Ro (*p* = 0.02 and *p* = 0.02, respectively) and anti-SSB/La antibodies (*p* = 0.01 and *p* = 0.001, respectively) were significantly higher in patients who registered an abnormal DLCO value. ANA positivity has also been linked to pulmonary manifestations in pSjD, independent of anti-SSA/Ro and anti-SSB/La positivity [34]. However, neither the RF (positivity rate) nor the ANA (titres or positivity rate) showed a notable difference between patients with low DLCO and those with normal DLCO values in our research.

In our study, multivariate analysis was performed for the aforementioned variables. Statistical significance was maintained for age at enrolment (*p* = 0.000), age at onset of the disease (*p* = 0.017), anti-SSA/Ro antibody positivity (*p* = 0.000), and anti-SSB/La antibody positivity (*p* = 0.000). The usefulness of anti-SSA/Ro antibodies, anti-SSB/La antibodies, and ANA titres in predicting a decreased DLCO in pSjD patients with interstitial changes in lung CT scans was evaluated by ROC analysis. The anti-SSB/La antibody titre was found to be the best predictor for low DLCO—AUC 0.791 (0.587–0.994), *p* = 0.016—in our patients.

When comparing patients with decreased or normal DLCO who concomitantly expressed hypocomplementemia, a *p*-value extremely close to the significance level was registered (*p* = 0.05). The result is not surprising since low levels of C3 have been previously associated with the presence of ILD in pSjD patients [30].

Patients with an NSIP pattern on HRCT were analyzed separately. The only factor that demonstrated a significant degree of association with this radiological pattern was renal involvement (*p* = 0.03). No similar findings could be found in the literature.

Both titres (*p* = 0.07) and positivity rate—*p* = 0.09, OR 0.15 (0.01–1.63)—of anti-SSB/La antibodies registered a close-to-significance decrease in patients with UIP pattern, suggesting a possible protective role of these autoantibodies against this type of ILD. Similar roles have been mentioned before in the literature for anti-SSA/Ro and anti-Ro52 antibodies [35], although these findings are inconsistent with the majority of studies on this matter. A close-to-significance *p*-value was also obtained for the male gender.

Patients diagnosed with LIP patterns on high-resolution CT scan images presented a significant association with the concomitant positivity for all four immunological markers that were searched (anti-SSA/Ro antibodies, anti-SSB/La antibodies, RF, and ANA)—*p* = 0.03—suggesting that quadruple positivity might represent a risk factor for developing LIP in pSjD patients. Once more, the determined difference in the anti-SSB/La antibody positivity rate was situated close to the level of significance (*p* = 0.09) when comparing these patients with the rest of the study population. This result aligns with the findings of Dong et al. [36] and Konak et al. [20], who identified higher anti-SSB/La antibody prevalence in LIP patients.

Without exception, the enrolled patients who presented an abnormal DLCO were associated other extraglandular involvement.

Quadruple positivity maintains its significantly increased frequency when comparing patients with LIP and abnormal DLCO with patients who registered a decrease in DLCO without associating an LIP aspect on the CT scan (*p* = 0.03).

Patients with low DLCO and NSIP radiologic patterns had close-to-significance *p*-values for renal impairment and hypergammaglobulinemia. The association of pulmonary manifestations in SjD with hypergammaglobulinemia was mentioned in previous studies [16].

In patients presenting low DLCO and UIP aspects in CT scan images, a significant increase in the enrolment age (*p* = 0.01) and TLC (*p* = 0.02) was noted when comparing them with the rest of the patients who registered abnormal DLCO values. Moreover, significantly inferior anti-SSA/Ro antibody titres were detected (*p* = 0.04), placing emphasis on the aforementioned idea that these immunological markers might play a protective role against UIP development. The difference in positivity rate for anti-SSB/La antibodies between patients with abnormal DLCO with or without UIP radiologic patterns was situated very close to the level of significance (*p* = 0.05). Seemingly, the presence of anti-SSB/La antibodies might have a protective role against this type of lung involvement, since a higher frequency was detected for patients without UIP.

The identification of patients with ILD and low DLCO detected at HRCT, who were not associated with restrictive disease, represents an argument that DLCO might be an early sign of lung involvement, and this can be used as a screening tool. However, pulmonary hypertension was not studied, as this could also be a potential cause of low DLCO.

The present study is limited by certain aspects. Firstly, HRCT was only performed in symptomatic patients. Secondly, the cross-sectional design of the research did not permit observing the evolution of lung lesions under the influence of the different studied factors. This is a significant aspect given the fact that UIP and NSIP are known to have the tendency of evolving towards progressive pulmonary fibrosis in pSjD patients [37]. In addition, the size of the study group was relatively small, and the number of male patients who were included was rather reduced. Finally, the detection of anti-Ro52 antibodies, serological markers that are known to be associated with ILD in SjD [24], could not be performed.

However, the association of an important number of clinical and para-clinical factors with ILD in SjD was investigated. In addition to anti-SSA/Ro antibodies, this study investigated the presence of other immunological factors, such as anti-SSB/La antibodies, ANA, and RF, which are not currently included in the classification criteria of SS. Moreover, both the titres and positivity rate of these autoantibodies were searched.

In conclusion, our findings reinforce the role of anti-SSA/Ro and anti-SSB/La antibodies in the pulmonary manifestations of pSjD. Their positivity rate was significantly associated with the decrease in DLCO in this research. Furthermore, although the role of anti-SSB/La antibodies as a predictive factor for ILD still requires clarification, the evidence suggesting this possibility is certainly growing. The presented results indicate anti-SSB/La antibodies as an important predictor of DLCO reduction and suggest a possible role in protecting against UIP development while favoring the evolution towards LIP. Thus, anti-SSB/La antibodies might represent a marker of lung involvement in pSjD patients. However, this hypothesis requires further research.

## 4. Materials and Methods

A retrospective study was performed. We studied consecutive pSjD patients presenting to the Internal Medicine Department and Rheumatology Department of Colentina Clinical Hospital, Bucharest, between 2019 and 2023, who underwent pulmonary functional tests due to the presence of respiratory symptoms, such as cough and dyspnea. Only those who were examined with HRCT and were proven to have ILD were included. The diagnostic work-up was completed with the measurement of DLCO in every case when ILD was detected on CT scan images. Patients presenting alternative diagnoses that had the potential to alter lung function were excluded from the study population. All participants were eighteen years of age or older.

Hematologic involvement was defined as the presence of anemia, thrombocytopenia, or leukopenia. Neurologic manifestations implied peripheral polyneuropathy, while renal impairment was considered in patients who registered changes in creatinine levels and renal clearance values or exhibited abnormal urinalysis findings.

The anonymity of all the data was ensured during the collection and analysis process. The Ethics Committee of Colentina University Hospital, Bucharest, approved the study protocol (No. 7/19 March 2025).

The enzyme-linked immunosorbent assay (ELISA) technique was used for the determination and quantification of the ANA, anti-SSA/Ro, and anti-SSB/La antibodies. Levels beneath 20 IU/mL were considered normal. For the RF, the cut-off was established at 14 IU/mL. RF, immunoglobulins, and the C3 and C4 complement components were determined by immunoturbidimetry. Hypocomplementemia was defined as either a decreased level of C3 (normal range 0.9–1.8 g/L) or C4 (normal range 0.1–0.4 g/L), while an increased level of immunoglobulin G (IgG, normal range 700–1600 mg/dL), immunoglobulin A (IgA, normal range 70–400 mg/dL), or immunoglobulin M (IgM, normal range 40–230 mg/dL) defined hypergammaglobulinemia.

Pulmonary function tests were performed with Quark PFT Cosmed (COSMED Srl, Rome, Italy) and included spirometry, lung volumes (forced vital capacity—FVC; total lung capacity—TLC), and diffusion tests. DLCO was presented as a percentage of the predicted value. A value higher than 80% was considered normal. Patients with values between 60 and 80% were categorized as having a mild decrease, while the results of 40 to 60% of the predicted value were defined as moderately decreased. A DLCO of 40% or less of the predicted value defined a severe decrease. Values below 80% were generally grouped under the terms “abnormal DLCO” or “low DLCO”. Patients with different radiological types of ILD were analyzed separately in order to define different patterns of patient characteristics.

Frequencies and mean ± standard deviations were used to express continuous variables with parametric distribution, and the means were compared by Student’s *t* test. The Mann–Whitney U test and Wilcoxon test were used to analyze the not normally distributed variables. Categorical variables were compared by the chi-square test. The odds ratio (ORs), 95% confidence intervals (CIs), receiver operating characteristic curves (ROCs), and area under the curve (AUC) were obtained. The hypotheses were evaluated using the two-tailed test. Statistical significance was defined by a *p*-value of less than 0.05. The collected data were statistically analyzed using the SPSS 20.0 software (IBM Corporation, Armonk, NY, USA).

## Figures and Tables

**Table 1 ijms-26-05867-t001:** Clinical and serological characteristics of the study group.

		Results
Gender (number of patients, percentage)	Male	4 (15.38%)
	Female	22 (84.61%)
DLCO (number of patients, percentage)	Normal	9 (34.61%)
	Abnormal	17 (65.38%)
FVC (mean L, % predicted)		3.08 (104%)
TLC (mean L, % predicted)		4.71 (92%)
Mean age at enrolment (years, mean ± standard deviation)		57.68 ± 11.77
Mean age at onset of disease (years, mean ± standard deviation)		51.38 ± 12.79
Mean duration of the disease (years, mean ± standard deviation)		7.04 ± 4.97
Serology—positivity (number of patients, percentage)	ANA	24 (92.30%)
	Anti-SSA/Ro antibodies	22 (84.61%)
	Anti-SSB/La antibodies	14 (53.84%)
	RF	14 (53.84%)
Serology—titres (UI/mL, mean ± standard deviation)	Anti-SSA/Ro antibodies	165.00 ± 90.75
	Anti-SSB/La antibodies	162.00 ± 49.46
	ANA	248.70 ± 188.53
NSIP radiologic pattern (number of patients, percentage)		15 (57.69%)
UIP radiologic pattern (number of patients, percentage)		5 (19.23%)
LIP radiologic pattern (number of patients, percentage)		6 (23.07%)
Cryoglobulinemia (number of patients, percentage)		3 (11.53%)
Hypergammaglobulinemia (number of patients, percentage)		9 (34.61%)
Hypocomplementemia (number of patients, percentage)		10 (38.46%)
Extraglandular involvement (except for ILD) (number of patients, percentage)		24 (92.30%)
Hematologic involvement (number of patients, percentage)		19 (73.07%)
Neurologic involvement (number of patients, percentage)		9 (34.61%)
Renal involvement (number of patients, percentage)		7 (26.92%)
Adenopathies (number of patients, percentage)		3 (11.53%)

DLCO—diffusing capacity of the lungs for carbon monoxide; ANA—antinuclear antibodies; RF—rheumatoid factor; NSIP—nonspecific interstitial pneumonia; UIP—usual interstitial pneumonia; LIP—lymphocytic interstitial pneumonia.

**Table 2 ijms-26-05867-t002:** Comparative analysis of abnormal and normal DLCO patients.

	Abnormal DLCO (17 Patients)	Normal DLCO (9 Patients)	OR (95% CI)	*p*-Values
Gender (female/male)	14/3	8/1	0.58 (0.05–6.58)	0.66
Mean age at enrolment (years, mean ± standard deviation)	61.88 ± 9.02	49.98 ± 12.73	-	0.02
Mean age at onset of disease (years, mean ± standard deviation)	55.24 ± 10.71	44.11 ± 13.82	-	0.03
Mean duration of the disease (years, mean ± standard deviation)	7.00 ± 5.00	7.11 ± 5.23	-	0.95
TLC (%, mean ± standard deviation)	0.92 ± 0.12	1.08 ± 0.14	-	0.01
Abnormal TLC (yes/no)	3/14	0/9	2.66 (0.25–27.48)	0.40
FVC (%, mean ± standard deviation)	0.84 ± 0.28	0.94 ± 0.08	-	0.18
Abnormal FVC (yes/no)	6/11	0/9	5.83 (0.61–55.74)	0.12
ANA titres (UI/mL, median values, min–max)	367.00 (12.00–508.00)	99.00 (5.00–509.00)	-	0.10
ANA (positive/negative)	17/0	7/2	6.75 (0.60–75.27)	0.12
RF (positive/negative)	10/7	4/5	1.78 (0.34–9.12)	0.48
Anti-SSA/Ro antibody titres (UI/mL, mean ± standard deviation)	105.93 ± 41.86	62.09 ± 51.06	-	0.02
Anti-SSA/Ro antibodies (positive/negative)	17/0	5/4	15.00 (1.44–155.31)	0.02
Anti-SSB/La antibody titres (UI/mL, median values, min–max)	62.00 (1.21–162.00)	2.66 (0.00–133.00)	-	0.01
Anti-SSB/La antibodies (positive/negative)	13/4	1/8	26.00 (2.45–275.88)	0.001
Quadruple positivity (RF+ANA+anti-SSA antibodies+anti-SSB antibodies) (positive/negative)	7/10	1/8	5.60 (0.56–55.42)	0.11
Extraglandular involvement (yes/no)	17/0	7/2	6.75 (0.60–75.27)	0.12
Hematologic involvement (yes/no)	12/5	7/2	0.68 (0.10–4.52)	0.69
Adenopathies (yes/no)	2/15	1/8	1.06 (0.08–13.65)	0.96
Neurologic involvement (yes/no)	7/10	2/7	2.45 (0.38–15.49)	0.33
Renal involvement (yes/no)	5/12	2/7	1.45 (0.22–9.61)	0.69
Smoking (yes/no)	2/15	1/8	1.06 (0.08–13.65)	0.96
Hypergammaglobulinemia (yes/no)	7/10	2/7	2.45 (0.38–15.49)	0.33
Hypocomplementemia (yes/no)	9/8	1/8	9.00 (0.91–88.57)	0.05
C3 titre (g/L, mean ± standard deviation)	1.01 ± 0.14	1.18 ± 0.33	-	0.17
C4 titre (g/L, median values, min–max)	0.00 (0.00–0.23)	0.00 (0.00–1.00)	-	0.26
Cryoglobulinemia (yes/no)	2/15	0/9	1.87 (0.17–20.60)	0.60
NSIP radiologic pattern (yes/no)	9/8	6/3	0.56 (0.10–3.02)	0.50
UIP radiologic pattern (yes/no)	3/14	2/7	0.75 (0.10–5.57)	0.77
LIP radiologic pattern (yes/no)	5/12	1/8	3.33 (0.32–34.12)	0.29

DLCO—diffusing capacity of the lungs for carbon monoxide; TLC—total lung capacity; FVC—forced vital capacity; ANA—antinuclear antibodies; RF—rheumatoid factor; C3—C3 serum complement component; C4—C4 serum complement component; NSIP—nonspecific interstitial pneumonia; UIP—usual interstitial pneumonia; LIP—lymphocytic interstitial pneumonia.

**Table 3 ijms-26-05867-t003:** Results of multivariate analysis of the abnormal and normal DLCO subgroups.

	Normal/Abnormal DLCO Univariate Analysis *p*-Values	Normal/Abnormal DLCO Multivariate Analysis *p*-Values
Age at enrolment	0.02	0.000
Age at onset of disease	0.03	0.017
Value of TLC	0.01	1.00
Anti-SSA/Ro antibody positivity	0.02	0.000
Anti-SSA/Ro antibody titres	0.02	1.00
Anti-SSB/La antibody positivity	0.005	0.000
Anti-SSB/Ro antibody titres	0.01	0.99
Hypocomplementemia	0.05	1.00

DLCO—diffusing capacity of the lungs for carbon monoxide; TLC—total lung capacity.

**Table 4 ijms-26-05867-t004:** Results of ROC analysis evaluating the utility of anti-SSA/Ro antibodies, anti-SSB/La antibodies, and ANA titres in predicting an abnormal DLCO.

	Area Under the Curve	*p*-Values	95% CI	
			Lower Limit	Upper Limit
Anti-SSA/Ro antibody titres	0.722	0.067	0.518	0.926
Anti-SSB/La antibody titres	0.791	0.016	0.587	0.994
ANA titres	0.699	0.100	0.470	0.929

ANA—antinuclear antibodies.

**Table 5 ijms-26-05867-t005:** Comparative analysis of patients with and without NSIP, UIP, and LIP radiological patterns.

	With NSIP Radiological Pattern (15 Patients)	Without NSIP Radiological Pattern (11 Patients)	OR (95% CI)	*p*-Values	With UIP Radiological Pattern (5 Patients)	Without UIP Radiological Pattern (21 Patients)	OR (95% CI)	*p*-Values	With LIP Radiological Pattern (6 Patients)	Without LIP Radiological Pattern (20 Patients)	OR (95% CI)	*p*-Values
Gender (female/male)	14/1	8/3	5.25 (0.46–59.25)	0.15	3/2	19/2	0.15 (0.01–1.58)	0.09	5/1	17/3	0.88 (0.07–10.46)	0.92
Mean age at enrolment (years, mean ± standard deviation)	55.93 ± 11.32	60.09 ± 12.48	-	0.38	58.60 ± 14.51	57.48 ± 11.43	-	0.85	61.33 ± 11.79	58.60 ± 11.84	-	0.39
Mean age at onset of disease (years, mean ± standard deviation)	50.07 ± 11.69	53.18 ± 14.55	-	0.55	50.60 ± 15.63	51.67 ± 12.47	-	0.88	55.33 ± 14.59	50.20 ± 12.34	-	0.40
Mean duration of the disease (years, mean ± standard deviation)	6.60 ± 4.45	7.64 ± 5.78	-	0.61	8.40 ± 5.22	5.71 ± 4.99	-	0.50	7.00 ± 6.63	7.00 ± 4.58	-	0.98
DLCO (percentage, mean ± standard deviation)	75.40 ± 18.40	79.27 ± 25.95	-	0.65	85.60 ± 36.03	75.00 ± 17.20	-	0.33	74.00 ± 15.32	77.95 ± 23.31	-	0.70
TLC (%, mean ± standard deviation)	0.97 ± 0.16	0.99 ± 0.12	-	0.70	3/2	14/7	0.75 (0.10–5.57)	0.77	5/1	12/8	3.33 (0.32–34.12)	0.29
Abnormal TLC (yes/no)	2/13	1/10	1.53 (0.12–19.47)	0.73	1.06 ± 0.13	0.96 ± 0.15	-	0.19	0.93 ± 0.10	0.99 ± 0.16	-	0.45
FVC (%, mean ± standard deviation)	0.88 ± 0.18	0.87 ± 0.30	-	0.85	0/5	3/18	0.79 (0.07–8.51)	0.84	1/5	2/18	1.80 (0.13–24.16)	0.65
Abnormal FVC (yes/no)	3/12	3/8	0.66 (0.10–4.17)	0.66	0.80 ± 0.45	0.89 ± 0.17	-	0.45	0.92 ± 0.13	0.86 ± 0.26	-	0.63
ANA titres (UI/mL, median values, min–max)	253.39 ± 176.34	242.30 ± 212.71	-	0.88	2/3	4/17	2.83 (0.34–23.01)	0.31	1/5	5/15	0.60 (0.05–6.44)	0.67
ANA (positive/negative)	14/1	10/1	1.40 (0.07–25.14)	0.81	204.81 ± 203.69	259.51 ± 188.51	-	0.57	273.54 ± 233.93	241.25 ± 179.20	-	0.72
RF (positive/negative)	8/7	6/5	0.95 (0.20–4.53)	0.95	5/0	19/2	0.90 (0.07–10.32)	0.93	5/1	19/1	0.26 (0.01–4.98)	0.34
Anti-SSA/Ro antibody titres (UI/mL, mean ± standard deviation)	93.64 ± 50.45	81.81 ± 49.27	-	0.73	2/3	12/19	0.5 (0.06–3.64)	0.49	4/2	10/10	2.00 (0.29–13.51)	0.47
Anti-SSA/Ro antibodies (positive/negative)	12/3	10/1	0.40 (0.03–4.47)	0.44	75.70 ± 42.03	94.34 ± 50.86	-	0.45	96.08 ± 56.70	88.15 ± 48.07	-	0.76
Anti-SSB/La antibody titres (UI/mL, median values, min–max)	50.00 (1.21–162.00)	7.00 (0.00–133.00)	-	0.54	5/0	17/4	1.66 (0.16–17.25)	0.66	5/1	17/3	0.88 (0.07–10.46)	0.92
Anti-SSB/La antibodies (positive/negative)	8/7	6/5	0.95 (0.20–4.53)	0.95	2.60 (0.00–63.00)	50.00 (1.08–162.00)	-	0.07	78.50 (1.08–133.00)	14.50 (0.00–162.00)	-	0.36
Quadruple positivity (RF+ANA+anti-SSA antibodies+anti-SSB antibodies) (positive/negative)	4/11	4/7	0.63 (0.11–3.41)	0.59	1/4	13/8	0.15 (0.01–1.63)	0.09	5/1	9/11	6.11 (0.60–62.23)	0.09
Extraglandular involvement (yes/no)	14/1	10/1	1.40 (0.07–25.14)	0.81	0/5	8/13	0.25 (0.02–2.52)	0.24	4/2	4/16	8.00 (1.06–60.32)	0.03
Hematologic involvement (yes/no)	11/4	8/3	1.03 (0.17–5.94)	0.97	4/1	20/1	0.20 (0.01–3.90)	0.25	6/0	18/2	1.10 (0.09–12.47)	0.93
Adenopathies (yes/no)	2/13	1/10	1.53 (0.12–19.47)	0.73	3/2	16/5	0.46 (0.06–3.64)	0.46	5/1	14/6	2.14 (0.20–22.47)	0.51
Neurologic involvement (yes/no)	5/10	4/7	0.87 (0.17–4.47)	0.87	1/4	2/19	2.37 (0.17–32.99)	0.51	0/6	3/17	0.64 (0.06–6.80)	0.71
Renal involvement (yes/no)	7/8	0/11	10.66 (1.12–101.34)	0.03	2/3	7/14	1.33 (0.17–9.91)	0.77	2/4	7/13	0.92 (0.13–6.39)	0.94
Smoking (yes/no)	2/13	1/10	1.53 (0.12–19.47)	0.73	0/5	7/14	0.31 (0.08–3.06)	0.31	0/6	7/13	0.25 (0.02–2.41)	0.23
Hypergammaglobulinemia (yes/no)	4/11	5/6	0.43 (0.08–2.26)	0.32	1/4	2/19	2.37 (0.17–32.99)	0.51	0/6	3/17	0.64 (0.06–6.80)	0.71
Hypocomplementemia (yes/no)	6/9	4/7	1.16 (0.23–5.80)	0.85	2/3	7/14	1.33 (0.17–9.91)	0.77	3/3	6/14	2.33 (0.36–15.05)	0.36
C3 titre (g/L, mean ± standard deviation)	1.10 ± 0.28	1.02 ± 0.14	-	0.39	0/5	10/11	0.18 (0.01–1.75)	0.14	4/2	6/14	4.66 (0.66–32.74)	0.10
C4 titre (g/L, median values, min–max)	0.00 (0.00–0.01)	0.00 (0.00–0.34)	-	0.64	1.03 ± 0.07	1.08 ± 0.26	-	0.66	1.02 ± 0.19	1.08 ± 0.24	-	0.50
Cryoglobulinemia (yes/no)	1/14	1/10	0.71 (0.04–12.82)	0.81	0.00 (0.00–0.34)	0.00 (0.00–1.00)	-	0.70	0.00 (0.00–0.30)	0.00 (0.00–1.00)	-	0.88

DLCO—diffusing capacity of the lungs for carbon monoxide; TLC—total lung capacity; FVC—forced vital capacity; ANA—antinuclear antibodies; RF—rheumatoid factor; C3—C3 serum complement component; C4—C4 serum complement component; NSIP—nonspecific interstitial pneumonia.

**Table 6 ijms-26-05867-t006:** Comparative analysis of patients with abnormal DLCO with and without LIP, UIP, and NSIP radiological patterns.

	With LIP Radiological Pattern (5 Patients)	Without LIP Radiological Pattern (12 Patients)	OR (95% CI)	*p*-Values	With UIP Radiological Pattern (3 Patients)	Without UIP Radiological Pattern (14 Patients)	OR (95% CI)	*p*-Values	With NSIP Radiological Pattern (9 Patients)	Without NSIP Radiological Pattern (8 Patients)	OR (95% CI)	*p*-Values
Gender (female/male)	4/1	10/2	0.80 (0.05–11.50)	0.87	2/1	12/2	0.33 (0.02–5.64)	0.43	8/1	6/2	2.66 (0.19–36.75)	0.45
Mean age at enrolment (years, mean ± standard deviation)	62.20 ± 12.96	61.75 ± 7.56	-	0.92	69.00 ± 2.64	60.36 ± 9.22	-	0.01	59.33 ± 7.12	64.75 ± 10.51	-	0.22
Mean age at onset of disease (years, mean ± standard deviation)	55.40 ± 16.42	55.17 ± 8.29	-	0.96	60.33 ± 11.54	54.14 ± 10.65	-	0.38	53.44 ± 6.93	57.25 ± 14.09	-	0.48
Mean duration of the disease (years, mean ± standard deviation)	8.00 ± 6.89	6.58 ± 4.29	-	0.61	5.33 ± 3.21	7.36 ± 5.32	-	0.42	7.00 ± 4.69	7.00 ± 5.65	-	1.00
TLC (%, mean ± standard deviation)	0.92 ± 0.10	0.92 ± 0.13	-	0.98	1.00 ± 0.00	0.90 ± 0.13	-	0.02	0.90 ± 0.15	0.95 ± 0.08	-	0.41
Abnormal TLC (yes/no)	1/4	2/10	1.25 (0.08–17.97)	0.87	0/3	3/11	0.75 (0.06–8.83)	0.81	2/7	1/7	2.00 (0.14–27.44)	0.60
FVC (%, mean ± standard deviation)	0.94 ± 0.13	0.80 ± 0.32	-	0.39	0.66 ± 0.57	0.88 ± 0.20	-	0.24	0.85 ± 0.23	0.83 ± 0.35	-	0.91
Abnormal FVC (yes/no)	1/4	5/7	0.35 (0.02–4.15)	0.39	2/1	4/10	5.00 (0.34–71.90)	0.21	3/6	3/5	0.83 (0.11–6.11)	0.85
DLCO (percentage, mean ± standard deviation)	69.80 ± 12.69	62.67 ± 10.99	-	0.26	69.80 ± 12.69	62.67 ± 10.99	-	0.26	63.33 ± 11.55	66.38 ± 12.22	-	0.60
ANA titres (UI/mL, median values, min–max)	325.08 ± 220.19	283.65 ± 173.16	-	0.68	125.00 (12.00–446.00)	404.69 (35.00–508.00)	-	0.19	313.42 ± 156.74	276.05 ± 216.25	-	0.68
RF (positive/negative)	4/1	6/6	4.00 (0.34–47.11)	0.25	1/2	9/5	0.27 (0.02–3.88)	0.32	5/4	5/3	0.75 (0.10–5.29)	0.77
Anti-SSA/Ro antibody titres (UI/mL, mean ± standard deviation)	115.06 ± 36.31	102.12 ± 44.90	-	0.54	63.00 ± 54.02	115.13 ± 34.52	-	0.04	115.17 ± 36.74	95.53 ± 48.09	-	0.36
Anti-SSB/La antibody titres (UI/mL, median values, min–max)	84.32 ± 56.78	56.85 ± 48.61	-	0.33	6.00 (2.00–63.00)	67.39 (1.21–162.00)	-	0.15	67.92 ± 49.07	61.57 ± 57.38	-	0.80
Anti-SSB/La antibodies (positive/negative)	5/0	8/4	3.33 (0.30–36.11)	0.32	1/2	12/2	0.08 (0.00–1.41)	0.05	7/2	6/2	1.16 (0.12–10.99)	0.89
Quadruple positivity (RF+ANA+anti-SSA antibodies+anti-SSB antibodies) (positive/negative)	4/1	3/9	12.00 (0.93–153.88)	0.03	0/3	7/7	0.25 (0.02–2.75)	0.25	3/6	4/4	0.50 (0.07–3.55)	0.48
Hematologic involvement (yes/no)	4/1	8/4	2.00 (0.16–24.32)	1.00	2/1	10/4	0.80 (0.05–11.50)	0.87	6/3	6/2	0.66 (0.08–5.53)	0.70
Adenopathies (yes/no)	0/5	2/10	0.61 (0.05–7.24)	0.69	0/3	2/12	1.08 (0.08–13.53)	1.08	2/7	0/8	3.87 (0.28–39.30)	0.33
Neurologic involvement (yes/no)	2/3	5/7	0.93 (0.11–7.82)	0.94	2/1	5/9	3.6 (0.25–50.33)	0.32	3/6	4/4	0.50 (0.07–3.55)	0.48
Renal involvement (yes/no)	0/5	5/7	0.22 (0.02–2.36)	0.21	0/3	5/9	0.41 (0.03–4.65)	0.47	5/4	0/8	10.8 (0.99–117.00)	0.05
Smoking (yes/no)	0/5	2/10	0.61 (0.05–7.24)	0.69	1/2	1/13	6.50 (0.28–151.12)	0.20	1/8	1/7	0.87 (0.04–16.74)	0.92
Hypergammaglobulinemia (yes/no)	3/2	4/8	3.00 (0.34–25.87)	0.35	2/1	5/9	3.60 (0.25–50.33)	0.32	2/7	5/3	0.17 (0.02–1.43)	0.09
Hypocomplementemia (yes/no)	4/1	5/7	5.60 (0.47–66.44)	0.14	0/3	9/5	0.15 (0.01–1.67)	0.12	5/4	4/4	1.25 (0.48–8.44)	0.81
C3 titre (g/L, mean ± standard deviation)	0.95 ± 0.10	1.04 ± 0.15	-	0.27	1.00 ± 0.00	1.01 ± 0.16	-	0.84	1.05 ± 0.18	0.97 ± 0.08	-	0.25
C4 titre (g/L, median values, min–max)	0.00 (0.00–0.01)	0.00 (0.00–0.23)	-	0.57		0.00 (0.00–0.23)	-	0.36	0.00 (0.00–0.01)	0.00 (0.00–0.23)	-	0.20
Cryoglobulinemia (yes/no)	1/4	1/11	2.75 (0.13–55.16)	0.49	0/3	2/12	1.08 (0.08–13.53)	0.95	1/8	1/7	0.87 (0.04–16.74)	0.92

DLCO—diffusing capacity of the lungs for carbon monoxide; TLC—total lung capacity; FVC—forced vital capacity; ANA—antinuclear antibodies; RF—rheumatoid factor; C3—C3 serum complement component; C4—C4 serum complement component; LIP—lymphocytic interstitial pneumonia; UIP—usual interstitial pneumonia; NSIP—nonspecific interstitial pneumonia.

## Data Availability

Data is contained within the article.

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
