# Peer review of "The Role of Anti-SSB/La Antibodies as Predictors of Decreased Diffusing Capacity of the Lungs for Carbon Monoxide (DLCO) in Primary Sjögren Disease"

_ijms, 2025, doi:10.3390/ijms26125867_

Round 1
Reviewer 1 Report
Comments and Suggestions for Authors
I have reviewed the submitted manuscript and found it quite interesting. It is a retrospective study that evaluated the autoantibody profile in a cohort of patients with Sjögren’s disease and its association with pulmonary involvement.
I would suggest the following points for improvement:
-
The authors should consider using the current terminology "Sjögren’s disease (SjD)" instead of "Sjögren’s syndrome."
-
It would be helpful to clearly state the total number of patients in follow-up, and among them, how many did not have pulmonary involvement. This would allow readers to better understand the frequency of pulmonary manifestations in the authors' cohort.
-
The tables should be improved. There are too many, and their current presentation may confuse the reader.
Reviewer 2 Report
Comments and Suggestions for Authors
The paper is interesting and well written. The authors correlated anti-SSA/Ro antibodies, anti-SSB/La antibodies, rheumatoid factor, antinuclear antibodies and diffusing capacity of the lungs for carbon monoxide (DLCO) in 26 primary Sjögren’s Syndrome patients who presented interstitial changes on the chest CT scan. The study demonstrated that anti-SSB/La positivity could be risk factors for developing Lymphocytic interstitial pneumonitis in primary Sjögren’s Syndrome patients and, thus, represent a marker of lung involvement in primary Sjögren’s Syndrome patients. The methodology is adequate and coerent with the endpoints. The results are well described and the discussion is coerent with the results and objectives. I sugges to discuss the role of IL-31/IL-33 axis and sHLA-G molecules in the development of chronic inflammatory immune-mediated diseases (see and add as reference the paper bu Murdaca et al concerning IL-31/IL-33 axis in chronic inflammatory immune-mediated diseases and the paper by Contini et al "sHLA-G molecules in immune mediated diseases and allergy" published in Frontiers in immunology).
Reviewer 3 Report
Comments and Suggestions for Authors
The article titled "The role of anti-SSB/La antibodies as predictors of decreased DLCO in primary Sjögren Syndrome" meets the criteria of an original research paper. The authors present clinically relevant and interesting findings regarding the potential role of anti-SSB/La antibodies as biomarkers for reduced diffusing capacity of the lungs (DLCO) in patients with primary Sjögren's syndrome (pSS).
Although the sample size is relatively small, it is acceptable given the rarity of pSS with interstitial lung involvement. The statistical analysis is clearly presented, and the use of multivariate and ROC analysis strengthens the reliability of the conclusions.
However, some aspects require clarification or further elaboration:
-
The use of the term “cross-sectional” raises some questions. Was this a truly cross-sectional study (based on a single time-point evaluation), or were data collected retrospectively or prospectively over a defined period?
-
It is not entirely clear if coexisting conditions that could affect lung function (e.g., cardiac disease, COPD) were also analyzed or excluded.
-
It would be helpful to specify the reference values for DLCO that were used to define “abnormal” or “low” results. The absence of this information makes it difficult to assess the clinical relevance of the findings.
Despite these minor concerns, the study offers a valuable contribution to the understanding of potential pulmonary biomarkers in Sjögren’s syndrome and is suitable for publication after minor revisions and clarifications.
